# A Multi-Level Path Analysis of the Relationships between the Momentary Experience Characteristics, Satisfaction with Urban Public Spaces, and Momentary- and Long-Term Subjective Wellbeing

**DOI:** 10.3390/ijerph16193621

**Published:** 2019-09-26

**Authors:** Minou Weijs-Perrée, Gamze Dane, Pauline van den Berg, Machiel van Dorst

**Affiliations:** 1Department of the Built Environment, Eindhoven University of Technology, Het Kranenveld 8, 5612 AZ Eindhoven, The Netherlands; g.z.dane@tue.nl (G.D.); p.e.w.v.d.berg@tue.nl (P.v.d.B.); 2Department of Architecture and the Built Environment, Delft University of Technology, Julianalaan 134, 2628 BL Delft, The Netherlands; m.j.vandorst@tudelft.nl

**Keywords:** Urban public spaces, experiences, satisfaction, subjective wellbeing (SWB), multi-level path analysis, geotagging, experience sampling method (ESM)

## Abstract

Previous research has shown that the urban environment could influence people’s behavior and wellbeing. However, little is still known about how the objective and subjective measures of the momentary experience of urban public spaces could contribute to the satisfaction with the urban environment of cities, which eventually could influence the momentary and long-term subjective wellbeing (SWB) of citizens. Therefore, the aim of this research is to gain insight into how momentary experience and satisfaction with the urban public space could contribute to the SWB of citizens, and thereby control for personal, contextual characteristics. Relationships were simultaneously analyzed using a multi-level path analysis approach based on a sample of 1056 momentary experiences of urban public spaces reported by 161 citizens of the urban area Eindhoven, The Netherlands. The results showed that personality and personal characteristics are highly important for explaining long-term SWB and subsequently long-term SWB positively influences momentary SWB (the degree of feeling secure, comfortable, happy and annoyed) together with the momentary satisfaction of urban public space characteristics. In addition, contextual characteristics, such as time/day and distance to facilities are important for explaining people’s momentary SWB. Policy makers and urban planners can use these results when developing policy and designing a healthy, attractive, livable and safe living environment for citizens.

## 1. Introduction

City-policy makers are increasingly suggesting that future “smart cities” should be citizen centric, smart and sustainable. These citizen-centered smart cities should focus both on social- and physical (natural environment, infrastructure, services, and buildings) processes in cities. It is also recognized that attractive and livable cities should be places where people feel safe, have positive experiences and are satisfied with their surroundings [1]. There already exists a large body of knowledge on the relation between people’s behavior and their environment in the field of environmental psychology [2]. Recently, there is also an increased scientific interest from the urban planning perspective on how the urban public spaces influences people’s subjective wellbeing (SWB) [3,4,5], as people are the most important actors in urban planning processes [6]. 

It is recognized that SWB can be conceptualized into momentary SWB (e.g., emotions/mood), as well as a stable long-term state (e.g., life satisfaction) [7,8]. Long-term SWB can be defined as “the degree to which an individual positively evaluates the overall quality of his/her life” [9]. Momentary SWB can be defined as the degree of feeling secure, comfortable, happy and annoyed at a specific moment in time [1]. Although urban planners and policymakers are also increasingly trying to create attractive public places that not only improve objective living conditions, but also to stimulate people’s SWB, there is still limited proof of which characteristics of momentary experiences of public places in the city could influence the momentary and long-term SWB of citizens. Urban public spaces can be described as spaces in the city that is accessible to the public, such as streets and parks, transportation facilities (e.g., train stations, airports or parking lots) or shopping facilities (e.g., supermarket or shopping mall) [10].

Previous studies have shown that urban public space could influence people’s overall quality of life or life satisfaction, which is a construct of long-term SWB [11,12,13]. However, measuring the overall quality of life might not be reflective of only the urban public space and planning as it consists of other conditions such as work and life-related issues. Besides the urban public space, previous research has shown that personal characteristics are also important indicators of long-term SWB [14,15]. Also, studies have shown that activities (e.g., daily activities, physical activities and leisure activities) are an important indicator for explaining long-term SWB [16,17,18]. However, these studies do not take the characteristics of momentary experiences, momentary satisfaction with the urban public space and momentary SWB into account. Therefore, to understand whether there is a direct influence of the perception of urban public spaces space on momentary and long-term SWB, people’s characteristics of momentary experiences should be measured. It is recognized that studies that take into account momentary user experiences of public spaces in real-world settings, are limited [19]. 

Only a few studies such as those by Dane [20], Birenboim [1], and Ettema and Smajic [4] discuss that characteristics of momentary experiences in urban public spaces affect people’s happiness (momentary SWB) and therefore probably also people’s long-term SWB. These studies, for example, showed that people who visit attractive places in the city with many activities going on (cafés, restaurants, shops, traffic) and in company (e.g., friends or family) are more likely to have positive momentary experiences that result in higher levels of (momentary) SWB. Also, in the field of environmental psychology, the benefits (e.g., stress reducing) of natural aspects (e.g., parks, public gardens or street trees) of urban public spaces for people’s SWB have been recognized [2,21]. Furthermore, transportation research found a relationship between people’s momentary feelings and momentary travel experiences [22,23]. However, these studies do not consider location characteristics of momentary experiences and are based on homogenous or small samples, limited time frames or restricted areas within a city.

Momentary experiences are expected to vary over space and time, depending on contextual variables, such as spatial characteristics (e.g., distance to facilities) or the weather at the moment of the momentary experience [24]. For example, when a person walks (leisure activity) through a well maintained green park during a sunny day at the weekend, this might lead to a positive momentary experience due to the objective characteristics (e.g., location, weekend day and no rain) and the subjective characteristics (e.g., satisfaction with the air quality, sounds of birds, and the clean green environment) of the urban public space. Eventually, such positive momentary experience might result in a positive affective state (e.g., feeling happy, comfortable and/or safe) that could be considered as momentary SWB, which subsequently could influence long-term SWB and vice versa.

Therefore, the aim of this research is to gain insight into which subjective and objective characteristics of momentary experiences of urban public spaces (experience characteristics) in cities could contribute to momentary and long-term SWB of citizens, and thereby controlling for personal characteristics. Whereas previous studies mainly focused specifically on momentary SWB, long-term SWB or on momentary subjective experiences, this study also takes into account the relations between these aspects. This study also contributes to previous research, by integrating objective characteristics through secondary data sources (e.g., the Royal Netherlands Meteorological Institute (KNMI) and OpenStreetMap (OSM)) to include more detailed information about the context, which is up until now still limited. 

Data was collected, during two weeks, using an experience sampling method (ESM) approach (a longitudinal research methodology to collect data at multiple occasions over time) to derive real-time data on positive and/or negative momentary experiences with regard to urban public spaces, the subjective perception of the environment (satisfaction with characteristics of the urban public space), which could give a more comprehensive understanding of momentary- and long-term SWB of citizens. A multi-level path analysis was used to analyze 1056 momentary experiences of urban public spaces reported by 161 citizens of Eindhoven, The Netherlands. 

The remainder of the paper is structured as follows. In the second section, the theoretical background is discussed. Next, the data collection and methodology are described, followed by a discussion of the main results. The final section contains the conclusion, limitations and recommendations for future work.

## 2. Literature Review

The level of attractiveness of cities, and pleasantness to live in cities depends on the subjective momentary experiences of individuals, such as an individual’s momentary and long-term SWB (e.g., level of happiness or life satisfaction) or satisfaction with locations (e.g., perceived safety or accessibility) in the city [1,13,19,25,26]. Two components of SWB can be described, namely affective wellbeing (AWB) (positive and negative emotions and moods) and cognitive wellbeing (CWB) (overall life satisfaction) [9]. It is recognized that people’s emotional state (momentary SWB) play an important role in shaping people’s satisfaction with the urban public space [27] and the other way around. It is expected that when people are more satisfied with their overall life (long-term SWB), they probably rate their momentary SWB as more positive. On the other hand, momentary SWB could also differ from people’s overall long-term SWB [1]. Thus, the following hypothesis is proposed:
**Hypothesis** **1** **(H1):**Momentary SWB is positively related to people’s long-term SWB.

With regard to personal characteristics, Argyle [14] suggested that almost 15%–20% of the variance in long-term SWB could be explained by demographic characteristics. For example, Cuñado and De Gracia [28] found a positive relation between education level and long-term SWB, as people higher educated have more confidence and opportunities for quality jobs and a higher income level. Furthermore, previous research found evidence that when age increases also life satisfaction increases, because older people spend more time on activities that could contribute to their momentary- and eventually to long-term SWB [29,30]. With regard to gender, previous studies showed conflicting results. Studies based on large international datasets showed that younger women experience higher levels of happiness than men and on the contrary older women experience lower levels of happiness than men [31,32]. Other studies showed that overall women have higher levels of life satisfaction than men [33,34,35]. However, some studies found no significant differences between men and women [36,37]. Argyle [14] found that people who have a job are overall happier (long-term SWB) than people who are unemployed. Moreover, Bloze and Skak [38] found that homeownership is positively related to long-term SWB. Another study showed that people who are married are more likely to be happier than people who are unmarried, separated or single [15]. In addition, Diener et al. [39] found a positive relation between marriage and life satisfaction. Saw, Lim and Carrasco [40] showed that health issues increases negative affect (AWB). Also, life changing events (e.g., marriage or death of a partner) could affect people’s feelings of happiness [41]. 

Furthermore, previous research showed that personality is an important indicator for long-term SWB [42]. For example, Gutiérrez et al. [42] showed that ‘neuroticism’ was the most strongly related to negative affect (AWB) and “extroversion” was the most strongly related to positive affect (AWB). In addition, Saw, Lim and Carrasco [40] found that personality (extroversion and emotional stability) could increase long-term SWB. Also, research showed that the personality traits extroversion, conscientiousness and agreeableness are positively related, and neuroticism negatively related to life satisfaction [43,44]. Overall, it is thus expected that personal characteristics (age, gender, education level, household composition, life-changing events, work situation and personality) are important for explaining long-term SWB. Therefore, the following hypothesis is formulated:
**Hypothesis** **2** **(H2):**Personal characteristics influence people’s long-term SWB.

Momentary experience of the urban environment, in this study, is based on respondents’ satisfaction with characteristics of the urban public space. It is recognized that for explaining satisfaction with regard to the urban public space, both spatial- and non-spatial factors are important [45]. For example, Poon and Shang [46] found that the perceived urban safety could have an influence on happiness, which is sometimes mentioned as a synonym for momentary SWB [9]. Attractiveness and openness of the environment are also important for explaining momentary SWB [47]. In addition, Dane [20] found that people that visit attractive places are more likely to have positive momentary experiences. Also, studies showed that air quality and noise levels in the environment could explain life satisfaction (long-term SWB) [48,49]. In addition, Bell, Greene, Fisher and Baum [2] showed that noise from public infrastructure can increase stress levels, which subsequently could affect momentary SWB. It is also recognized that smell influences our momentary experiences of urban places [50]. Furthermore, it is important to have nearby green areas to get distressed and relaxed and to decrease noise levels [51]. Moreover, Bakolis et al. [52] found relationships between natural elements of the environment, such as seeing trees, hearing birds singing, seeing the sky, and feeling in contact and people’s momentary mental well-being. Also, Kaplan [53], based on the “Attention Restoration Theory”, argued that being in contact with the natural environment increases the opportunity to de-stress. Overall, it is expected that people’s positive perception of urban characteristics leads to momentary satisfaction with the urban public space, which subsequently might increase momentary and long-term SWB. Based on the above described literature, the following hypothesis is proposed:
**Hypothesis** **3** **(H3):**People’s momentary satisfaction with characteristics of the urban public space (e.g., urban safety, natural elements, air quality, aesthetic quality, smell, accessibility and noise) is positively related to momentary and long-term SWB.

According to the activity theory, it is suggested that people are happier with their life (long-term SWB), if they are involved in interesting and engaging activities [26,54]. Previous studies showed that activities (e.g., daily activities, physical activities and leisure activities) are an important indicator for explaining momentary- and long-term SWB [16,17,18]. For example, Saw, Lim and Carrasco [40] found that the level of physical activity positively influences positive affect (AWB). In addition, not only the type of activity, but also the company (together with for example acquaintances or alone) is expected to have an influence on the perceived momentary SWB [4,55]. Moreover, travel behavior (e.g., travel mode) has been found to influence SWB. For example, Ettema et al. [22] developed the satisfaction with travel scale (STS), which consisted of both affective and cognitive components related to travel experiences. They found that using a car is most positively related to mood (affective momentary SWB). In addition, the results showed that STS and mood are positively related. Other studies found that the more active travel modes (e.g., walking and bicycling) are more likely to be positively evaluated than using a car or public transport [56,57]. Another study found that people’s positive mood (momentary SWB) is higher when walking and bicycling (Glasgow et al., 2019) [58]. Also, previous knowledge of a place could influence people’s momentary SWB ( the degree of feeling secure, comfortable, happy and annoyed) [1]. 

**Hypothesis** **4** **(H4):**
*Momentary experience characteristics are likely to influence momentary SWB.*


Hypothesis 4a (H4a): Interesting, engaging and physical activities during a momentary experience are positively related to momentary SWB.Hypothesis 4b (H4b): Company during a momentary experience is positively related to momentary SWB.Hypothesis 4c (H4c): Active travel behavior of a momentary experience is positively related to momentary SWB.Hypothesis 4d (H4d): Familiarity with an urban space during a momentary experience is positively related to momentary SWB.

With regard to contextual (objective) characteristics, research showed that weather characteristics are also important for explaining momentary SWB. For example, Tsutsui [59] found that, based on data on daily events, feelings of happiness are more affected by the temperature during the events than by the overall average temperature of a day. This study did not find any significant effect of humidity, wind speed, precipitation, or sunshine. Furthermore, day of the week and time of day are important influences on subjective momentary SWB. For example, people report more positive feelings at the weekends [1,54]. Finally, the shorter the distance to, and accessibility of facilities (e.g., public transport and shopping facilities), could positively affect life satisfaction (long-term SWB) [12,13,60]. For example, walking distance to public transport stop was found to be negatively related to life satisfaction [12]. However, these studies focused on the distance related to their living environment. This current study focuses on experiences in the city. Therefore, it is expected that the distance to main facilities of the experience location not only influences long-term SWB, but also momentary satisfaction with the urban public space and momentary SWB.

**Hypothesis** **5** **(H5):**
*Contextual characteristics of momentary experiences are likely to influence people’s momentary satisfaction with the urban public space and SWB.*


Hypothesis 5a (H5a): Good weather positively influences people’s momentary satisfaction with the urban public space and SWB. Hypothesis 5b (H5b): Experiences that take place at the weekend are positively related to momentary SWB.Hypothesis 5c (H5c): Distance to facilities in the city influences people’s momentary satisfaction with the urban public space and SWB.

Overall, it is assumed that momentary experience characteristics (activity type, company, familiarity of the location) influence momentary SWB. In addition, contextual characteristics such as weather and time/day could influence how satisfied people are with the urban public space in time and space (momentary satisfaction with characteristics of urban public spaces, such as air quality, noise, natural elements, smell or accessibility) and SWB. Finally, it is hypothesized that momentary SWB could influence the long-term SWB and vice versa. Figure 1 showed the hypothesized relationships in a conceptual model.

## 3. Materials and Methods 

### 3.1. Data Collection 

The data collection instrument consisted of two parts. The first part was an online questionnaire, which consisted of multiple-choice questions about relevant demographics, work situation, personality, health and long-term SWB. Participants were also asked to indicate whether they were willing to participate in the second part of the research and were therefore asked to report their email address. With regard to the second part, a web-based ESM questionnaire was used. ESM is a useful method to gather information about momentary experiences immediately after they occur. There are three types of ESM, namely signal contingent, interval contingent and event contingent [61,62,63]. In the signal contingent method, participants are prompted (e.g., with smartphones devices) at random times within a fixed time period to report their experience or activity [64]. On the contrary, in the interval contingent method, participants report their events at predetermined intervals (e.g., every hour or daily) [65]. For the event contingent method, participants need to report all events at the moment when they occur [66]. In this study, the event-contingent method was to collect data on all momentary experiences in the city. As the number of negative and positive momentary experiences, with regard to the urban public space, will be limited, the event-contingent method was recommended [58]. 

In the ESM questionnaire, respondents were asked to report all their positive and/or negative momentary experiences with regard to any urban public locations/places (e.g., park, square, city center, shopping center) within the urban area of Eindhoven, for two weeks in June. For example, when participants walked at the train station and they did not feel safe or when they were shopping in the city center and they felt very happy or satisfied about their surroundings, they could geotag their location on the survey and answer the questions related to the satisfaction with the urban public space. To remind the participants to report all their momentary experiences, they received two email reminders every day, with the link to the online questionnaire. Before the ESM research period, they received an instruction with examples of positive/negative momentary experiences. It was explained that only the experiences related to urban public spaces, excluding experiences at home and/or at the work environment, should be reported.

Data was collected in June 2019 among citizens of the municipality of Eindhoven, which is one of the five largest cities in The Netherlands and part of the Metropolitan Region Brainport Eindhoven. The link to the web-based questionnaire of the first part of the research was distributed by a newsletter of the municipality among citizens who are registered to be willing to participate in research conducted by the municipality of Eindhoven. Citizens that were willing to participate in the first and second part of the research received a gift voucher of 10 euros. Because of the limited budget, the first part of the questionnaire was restricted to 300 participants. Of these 300 participants, 161 reported 1056 (relevant) momentary experiences in the second part of this study. These momentary experiences were used in the analyses. On average, participants reported 6.56 momentary experiences over two weeks, with a minimum of 1 and a maximum of 32 momentary experiences and a standard deviation of 6.32. Figure 2 shows the distribution of the number of momentary experiences reported per participant. Although respondents received reminders every day, the response on event-level is still low. Many participants only reported one momentary experience in Eindhoven during two weeks, which could result in a biased sample. Therefore, an exploratory *t*-test was used for analyzing differences, with regard to momentary SWB and satisfaction with urban public spaces between two groups, namely (1) people who reported only one experience and (2) people who reported more than one experience. However, no significant difference was found between these two groups. Thus, based on the sample of 1056 momentary experiences in the city, relationships between the momentary satisfaction with urban public spaces, momentary- and long-term SWB could be analyzed.

### 3.2. Measures

#### 3.2.1. Personal Characteristics

Participants were asked about their age, gender, income, education level, household composition, homeownership, health, work situation and life-changing events using multiple-choice questions. In addition, personality was measured using the 10-item scale BFI-10 by Rammstedt and John [67] to measure the Big Five personality traits (extroversion, agreeableness, conscientiousness, neuroticism and openness). This short scale is based on the 44-item Big Five Inventory [68]. Finally, self-reported health was measured by asking respondents ‘How would you say that, in general, your health is?’ with responses on a 5-point Likert scale ranging from excellent to poor. It is recognized that measuring health with a single question is a valid and reliable indicator of health [69]. For the analysis, the categories of the personal categorical variables were recoded into dummy variables.

#### 3.2.2. Long-Term Subjective Wellbeing (SWB)

To measure long-term SWB, a similar measurement of Saw, Lim and Carrasco [40] was used. They measured SWB, using two items, namely the first item was based on the Satisfaction with Life Scale (SWLS) to measure satisfaction with life [70] and the second item was based on the 10-item international Positive and Negative Affect Schedule (PANAS) Short Form (I-PANAS-SF) [71]. Items used for positive affect were inspired, alert, attentive, active, and determined. For negative affect, the items were afraid, upset, nervous, ashamed and hostile. These items were measured using a 5-point Likert scale, ranging from never to always. Overall, the following sum was used to measure long-term SWB: long-term SWB = positive affect – negative affect + life satisfaction. The Cronbach’s Alpha of the total sum of all 15 items, namely five positive affect items, five negative affect (reversed scored) items and five items of satisfaction with life, is 0.790. This is enough to sum the scores to measure long-term SWB.

#### 3.2.3. Momentary SWB

To analyze people’s momentary SWB, respondents were asked to rate how secure, comfortable, happy and annoyed they felt at the moment they reported their momentary experiences, on a 5-point Likert scale (adapted from Birenboim [1]). As these emotions correlate with each other, the sum score was used in the analyses. The Cronbach’s Alpha for the total score is 0.872, which is high enough to sum the scores of these emotions for the analyses.

#### 3.2.4. Momentary Satisfaction with Urban Public Space

Respondents were asked to indicate to what extent they were satisfied with the characteristics ( air quality, aesthetic quality, atmosphere, smell, accessibility, sufficient parking spaces, distance to facilities, traffic safety, natural elements, noise and cleanliness and maintenance of the space) of the location of and at the time of their momentary experience, on a 5-point Likert scale, ranging from very dissatisfied to very satisfied. To limit the number of parameters in the analyses, the sum score of the satisfaction with all these 12 items was used. The Cronbach’s Alpha of 0.889 shows a good internal consistency of the items.

#### 3.2.5. Momentary Experience Characteristics

First, participants were asked to give the name and to indicate on an interactive map of Eindhoven, by dragging and dropping the pin on the map or using Global Positioning System (GPS)-location of their device, of the exact location of their momentary experience. Subsequently, they were asked about the time of the momentary experience, whether the location of their momentary experience was indoors or outdoors, the type of location (e.g., work, shop/mall etc.) and their main activity at this location. In addition, they were asked about the main transportation mode they used to travel to the location. Furthermore, it was expected that previous knowledge of a place (familiarity) could influence momentary SWB [1]. Therefore, respondents were also asked about their familiarity with the location, based on a 5-point Likert scale. As it is recognized that having company during an activity could positively influence how people momentary experience places in the city and momentary SWB [4,55], participants were also asked whether they were alone, together with one other person or together with two or more people.

Based on the location (using geographic coordinates of the location of the momentary experience), time and date of the momentary experiences that were reported, several additional situational variables were collected such as location data (distance to facilities), which was extracted from the OSM database of Eindhoven city, and weather data (temperature, the occurrence of rain and the cloudiness), which was extracted from KNMI. The distance (in kilometers) to main facilities in the city, namely shops, restaurants and public transport stops was used in the analyses. These facilities were chosen, because previous research showed that the distance to these facilities could influence people’s SWB [12,13,60]. With regard to weather, KNMI measured temperature in 0.1 °C at 1.50 m height every hour. Cloudiness was measured, every hour, based on a scale from 1 to 9, whereby 9 means that the upper air is invisible. Finally, rain was measured based on whether rain occurred (1) or not (0) during the observation (every hour).

### 3.3. Analytical Approach

To simultaneously analyze the effects of the urban public space, personal- and contextual characteristics on people’s SWB, as well as the relationships between the dependent variables, namely satisfaction of the urban public space, long-term and momentary SWB, a multi-level path analysis approach was used. Path analysis is an extension of a multiple regression analysis and a special form of structural equation modelling that only includes observed or measured variables (Streiner, 2005) [72]. This method is able to simultaneously analyze relationships between several dependent and independent variables, which is required for this study. Another of the advantages of a multi-level path model is that this approach is able to take into account the hierarchical structure of the data (multiple momentary experiences per participant, which have to be treated as clusters) [73]. The multi-level path model was estimated using the statistical software package LISREL version 8.54 [74]. See Hox and Roberts [75] for a more in-depth explanation of multi-level models. 

A two-level model was estimated, which allows for variation at the within-level (level of momentary experiences) and the between-level (participant level) and which includes residuals at both levels. The cluster effects represent unobserved participant characteristics that could affect the outcomes for momentary experiences. Personal characteristics (e.g., gender, age and personality) and long-term SWB are used to explain the between-level patterns. The momentary experience characteristics (e.g., activity type, location type, weather, day of the week, distance to facilities, momentary SWB and satisfaction location characteristics) are used to explain the variance at the within level.

To build the model structure, a stepwise model selection was used by adding independent variables (categorical variables were recoded into dummy variables) and their relationships that were expected to have a relationship based on the literature review and found to be significant using bivariate analyses. Correlations of personal characteristics and experience characteristics are allowed in the model [73]. To reduce the number of variables in the path analysis and not overfitting the model, the links that were not found to be significant in the path model, at the 0.05 significance level (*t* ≥ 1.96), were subsequently removed. This common backward stepwise process (see e.g., [73]) was repeated until a model was estimated with only significant relationships and the best model fit.

## 4. Results 

### 4.1. Participants and their Momentary Experiences

Table 1 shows the main characteristics of the sample. The sample consists of a comparable share of women and men. As can be seen, the sample is not completely representative of the population of Eindhoven, as it contains a higher percentage of people aged over 45 years and a lower percentage of people aged 35 years or younger. This is probably also related to the high percentage of people who are retired (26%) and, therefore, have more time to participate. In addition, a high percentage works full-time (35%) or part-time (19%). Also, people who are living alone and tenants are underrepresented in this sample. Most participants rate their health as good, excellent or very good.

As can be seen in Table 2, most momentary experiences took place outdoors and approximately half of the momentary experiences were alone. Momentary experiences took place most frequently on the road or when relocating (37%), at a public outdoor space (21%), or at a shop/mall (14%). The average temperature during the momentary experiences was approximately 24 °C and during 5% of the momentary experiences rain occurred. Most momentary experiences took place near shops (mean = 0.27 km), restaurants (mean = 0.35 km) and public transport stops (mean = 0.20 km).

### 4.2. Multilevel Path Model

Table 3 shows several measures of model fit. Rules of thumb suggest that a model provides a good fit of the data if the value of Chi Square divided by the degrees of freedom is close to 1 [76]. In addition, the root mean square error of approximation (RMSEA) should preferably be lower than 0.05 [77]. Overall, it can be concluded that the model shows a good fit with the data. Figure 3 shows the final estimated path model with only the direct standardized significant effects at within level (momentary experience level) and at between level (participant level). The R-squares in Table 4 show that the explanatory power of the independent variables, related to the participant, is relatively strong for long-term SWB. 

### 4.3. Participant Level 

As expected, the results show a positive effect of long-term SWB on momentary SWB, which suggests that people who are overall more satisfied with their life and are more positive, also rate their momentary SWB (happy, secure, comfortable, and annoyed) higher. No effect was found the other way around (of momentary SWB on long-term SWB). Thus, H1 is partly confirmed. 

With regard to the work situation, the results showed that people who are retired have a higher perceived long-term SWB. Furthermore, people who own their home perceive their long-term SWB to be higher than people who rent their home. People who rate their health as reasonable/bad are also more likely to perceive their long-term SWB lower. Thus, H2f is confirmed. Moreover, the results showed that when people moved to another dwelling over the past year (life-changing event), they rate their long-term SWB higher. Long-term SWB was found to be influenced by personality traits, namely extroversion was found to have a positive effect and neuroticism a negative effect on long-term SWB. These results suggest that people who are more extroverted (more outgoing and sociable) score higher on long-term SWB. On the other hand, people who are more neurotic and anxious are more likely to rate their long-term SWB lower than people who are less neurotic. This current study did not show any significant effects of age, gender and education level on long-term SWB. Overall, the results show that personal characteristics are important for explaining long-term SWB and should, therefore, be included in future research on the relationship between urban public spaces and momentary- and long-term SWB. 

### 4.4. Momentary Experience Level

At the within level (the level of momentary experiences), the degree of satisfaction with the urban public space was found to positively influence people’s momentary SWB. This result suggested that people who are more satisfied with their surrounding are more likely to rate their momentary SWB higher. Based on this result, H3 is confirmed.

Furthermore, significant relationships were found between the location of the momentary experience and momentary SWB. The location leisure/café/bar/restaurant was found to be positively related to momentary SWB. A shop/mall and when relocating were found to be negatively related to momentary SWB. Thus, H4a is accepted. With regard to the transportation mode (H4c), having company (H4b) and familiarity with the place (H4d) the results show no significant relationships with momentary SWB. Therefore, these hypotheses are rejected.

Next, contextual characteristics were found to affect people’s momentary SWB. When the momentary experience took place at the weekend, respondents recorded higher momentary SWB. This finding confirms H5a. Somewhat unexpected, positive relationships were found between the distance of shops and public transport stops and momentary SWB. So, H5c is accepted. No significant effect was found of weather on momentary satisfaction with the urban public space and momentary SWB. Thus, H5b is rejected. 

## 5. Discussion and Future Research

The main aim of this research was to analyze the relationships between the momentary satisfaction with the urban public space, momentary- and long-term SWB and how these concepts are influenced by personal characteristics, characteristics of positive or negative momentary experiences in the city and contextual characteristics of these momentary experiences. The results showed a relationship between the overall satisfaction with the urban public space and momentary SWB. This result confirms the importance of the urban public space for people’s wellbeing. Previous research also showed that safety, attractiveness of a place, air quality, noise, smell and natural elements of the urban public space could affect momentary and long-term SWB [9,20,46,48,49,50,52]. Future research could look in more detail at the relationships between satisfaction with characteristics of the urban public space (air quality, aesthetic quality, atmosphere, smell, accessibility, sufficient parking spaces, distance to facilities, traffic safety, natural elements, noise and cleanliness and maintenance of the space) and SWB.

Furthermore, the results showed that the type of location, namely a leisure/social location (e.g., café/restaurant) (positive) shopping locations (negative) and relocating locations (negative) are important for explaining momentary SWB. This is also recognized by previous studies that showed that leisure/social locations are locations where interesting and engaging activities take place that could increase momentary SWB [26,54], compared to locations where less interesting activities (e.g., relocating/travelling or grocery shopping) take place. Glasgow et al. 2019 [58] also found that during errand trips (travelling to post office, grocery store or to the doctor) people’s mood is more negative compared to other trips. Also, people are more satisfied about their urban public space outside of the city center (longer distance to shops and public transport stops). This result is probably related to the higher level of natural features outside of the city center that contributes to momentary SWB [20]. Also, leisure- and public outdoor locations are more likely to be located outside of the city center. Previous studies showed that common activities at such locations (e.g., physical activities and leisure activities) are important indicators for explaining also the long-term SWB [16,17,18]. It is also confirmed by previous studies that people have more positive momentary experiences at the weekend [1,54]. Probably, the above described activities at leisure/social locations take place more often at the weekend, when people are less likely have to work. These results could help policymakers and urban planners to create more attractive and pleasant urban public spaces, by focusing on creating more attractive public locations (e.g., related to leisure or social meeting places) in the city center, with more natural features (e.g., green and water elements), that contribute to momentary SWB. It is also recognized that urban spaces that are most preferred are well organized, but also contain diversity to stay interesting for the users [45].

Similar to existing theory, this study provided evidence for the importance of people’s personality for explaining long-term SWB. More extroverted people rate their long-term SWB higher and the more neurotic and anxious people rate their long-term SWB. The importance of personality, in particular the personality traits extroversion and neuroticism, for explaining SWB was also confirmed by previous research [38,40,43]. Furthermore, the results on the relationship between work situation and long-term SWB suggests that retired people have more time to spend on positive activities that could contribute to their long-term SWB, which was also confirmed by previous research [29,30]. With regard to homeownership, similar results were found by Bloze & Skak [38]. This finding could be used by policy makers for developing housing policy that focusses on stimulating homeownership. As expected, people with lower health conditions rate their long-term SWB lower. This relation was also found by Saw, Lim & Carrasco [40]. Furthermore, the finding on the life event moving contradicts the findings by Luhmann [78], who suggest that moving is a stressful event, because people have to adjust to their new situation. On the other hand, people could also move to a house that better fits their needs and preferences, which might result in a higher SWB.

Although previous studies found several effects of age [30], gender [32] and education level [28], this current study did not show any significant effects on long-term SWB. Thus, other personal variables (e.g., personality, health, life changing events, home ownership and work situation) were found to be more important for explaining long-term SWB. Also, it is probably caused by the fact that several characteristics of momentary experiences were included in the model.

## 6. Conclusions 

Research that analyses the relationships between people’s emotions ( momentary SWB), long-term SWB, and how people perceive urban public spaces, is still limited [79]. Therefore, the main contribution of this study is that it takes into account the relationships between momentary SWB, long-term SWB and momentary satisfaction in an empirical study; whereas previous studies mainly focus on only one of these concepts. Momentary satisfaction was measured based on respondents’ satisfaction with characteristics of the urban public space (e.g., air quality, aesthetic quality, atmosphere, smell, accessibility and noise), which was not considered in other studies that dealt with momentary experiences. Moreover, this study contributes to existing theory by analyzing all expected relationships between momentary satisfaction with the urban public space, momentary- and long-term SWB simultaneously in a single model and controlling for several personal, contextual and momentary experience characteristics, using a multi-level path analysis. In addition, previous studies on momentary SWB are mainly based on a homogeneous sample (students) [1,4]. This study used an ESM approach to derive a more heterogeneous sample among citizens of Eindhoven. This helped to get more insights into how momentary SWB could vary over time and place. Research that combines subjective and objective measurements is still limited, but highly important to get a better understanding of people’s feelings and momentary experiences of the urban public space [80]. Therefore, another contribution is that this current study not only takes into account subjective measures, but also includes several objective variables (weather, distance to facilities), based on secondary location data, derived from OSM and KNMI. 

Although using ESM has several advantages, this approach also led to several limitations. Because of the time and commitment that was demanded from participants by using an ESM approach led to a low response rate for the second part of the study compared to the first part. Using a more representative and larger sample, also from other cities and counties, could increase the interpretation and generalizability of the results. Furthermore, respondents were asked to report positive and negative momentary experiences at any urban place in the city. Many respondents reported only one momentary experience, which suggests a high rate of non-response at the event level. Therefore, it was not possible to look at the number of momentary experiences at the urban area and whether people have more positive momentary experiences in certain areas compared to other areas in the city.

Due to the developments in ICT in recent years, it is now possible to exploit dynamic measurement tools, such as near-real time surveys and geotagging information in studies. Such data collection methodologies enable to collect data from large samples and to analyze momentary experiences. This study implemented a (near-real time) web-based ESM survey with a geotagging functionality that was also accessible by a mobile phone. However, using a smartphone application with a more user-friendly environment could help to obtain a higher response rate. In addition, as the momentary experiences and satisfaction are relatively new subjects in urban planning studies, this study can contribute to a standardization of data collection methodologies for future research.

With the novel approach used in this current study, it is possible to gain an insight into where the experiences took place which can be used to extract more data about surroundings that can help to develop policy on health and wellbeing in urban areas. However, the objective measures related to facilities used in this study are limited compared to the data from OSM which is built by a community of mappers that contribute and maintain data about roads, trails, cafés, railway stations, etc. Therefore, including more objective measures of these characteristics would be interesting for future studies. Also, including the number of facilities in the immediate environment (100 meter radius of the momentary experience location) instead of the closest distance to shops, café’s restaurants and public transport stops, could show different results. 

Overall, this study provides new insights into how the momentary experience of urban public spaces contributes to the satisfaction with the urban public space of cities and momentary and long-term SWB of citizens. For example, satisfaction with the overall urban public space was found to be important for explaining momentary SWB, which was not considered in previous research. Future more in-depth research is needed on unique aspects of urban public spaces and how these relate to the wellbeing of citizen.

## Figures and Tables

**Figure 1 ijerph-16-03621-f001:**
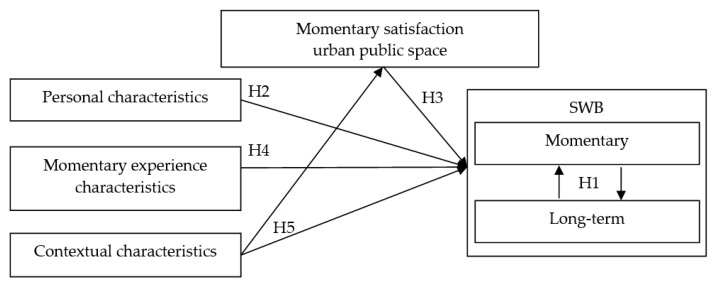
Conceptual model.

**Figure 2 ijerph-16-03621-f002:**
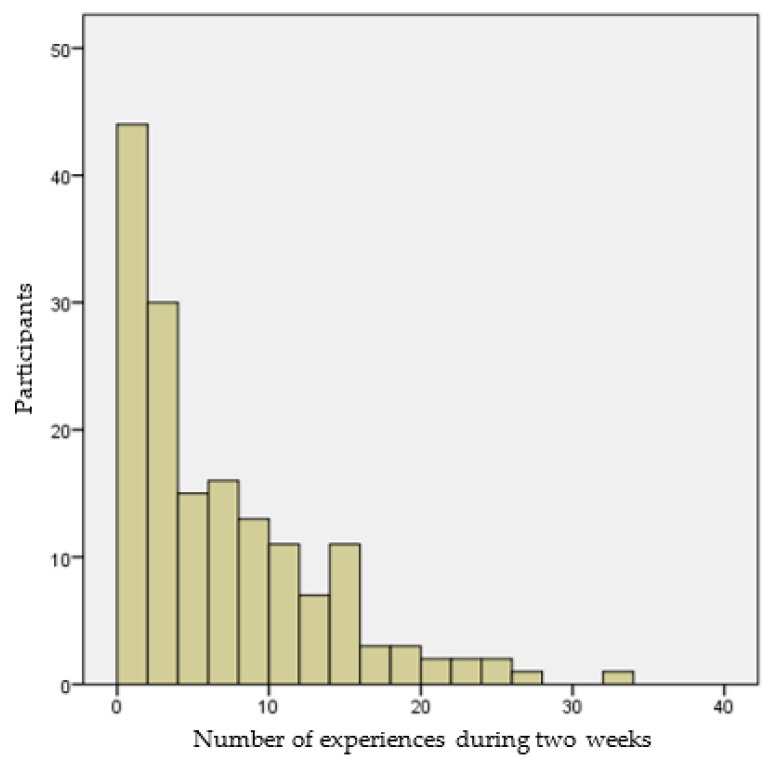
Distribution of the number of experiences per respondent.

**Figure 3 ijerph-16-03621-f003:**
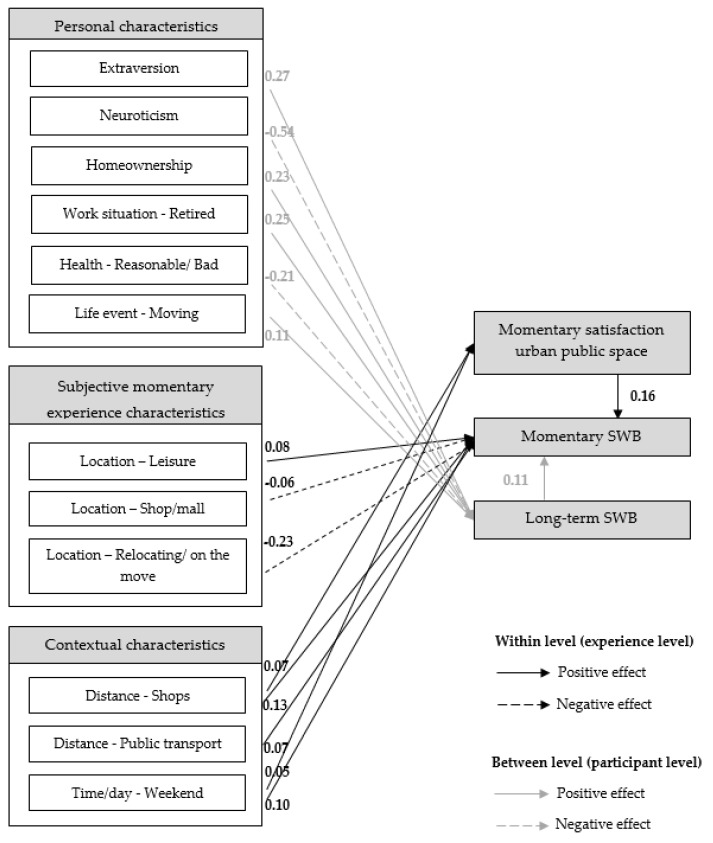
Significant standardized effects.

**Table 1 ijerph-16-03621-t001:** Participants characteristics (*n* = 161).

Gender	Sample (*n*)	Sample (%)	Eindhoven 2019 (%)
Male	84	52	51
Female	77	48	49
Age			(>20 years)
Age (18–35 years)	17	11	32
Age (35–45 years) Age (46–55 years) Age (56–65 years)	26 39 39	16 24 24	17 16 15
Age (>65 years)	40	25	21
Household composition			
One-person household	45	28	48
Couple without children	73	45	25
Couple with children	35	22	26 (households with children and other)
Single parent family and other	8	5	
Work situation			
Self-employed	12	7	64 (percentage of people with income from work (>12 h)
Full-time	56	35	
Part-time	30	19	
Unemployed Retired	19 42	13 26	36
Education			
Low education level	51	33	NA
Medium education level	70	43	
Higher education level	38	24	
Homeownership			
Home owner	127	79	47
Tenant	34	21	53
Health			
Reasonable/Bad Good Excellent/Very good	26 69 66	16 43 41	NA
	Mean	St. deviation	
Long-term subjective wellbeing (SWB)	26.46	5.520	
Life satisfaction	18.13	3.548	
Negative affect	10.83	2.093	
Positive affect	19.17	1.725	
Personality traits			
Extroversion	6.52	1.803	
Agreeableness	7.84	1.212	
Conscientiousness	7.66	1.341	
Neuroticism	4.78	1.544	
Openness	7.47	1.565	

**Table 2 ijerph-16-03621-t002:** Momentary experience characteristics (*n* = 1056).

Indoor/Outdoor	Sample (*n*)	Sample (%)
Indoor	174	17
Outdoor	882	83
Company		
Alone	612	58
One or more other people	444	42
Location type		
Work	44	4
On the road(relocating)	386	37
Shop/mall	148	14
Café/bar/restaurant Culture/sport venue/facility Public outdoor space (e.g., park) Other	72 57 219 130	7 6 21 12
Activity type		
Work/studying Travelling/relocating Social meeting (informal) Shopping Leisure/cultural activity Other	69 405 142 186 77 172	7 38 12 18 7 16
Transportation mode		
Car	192	18
Bicycle	511	49
Walking	320	30
Public transport Other	21 12	2 1
	Mean	St. deviation
Familiarity	4.58	0.693
Satisfaction urban public space	41.22	7.824
Momentary SWB	3.91	1.058
Location characteristics in kilometers (extracted from OpenStreetMap (OSM))		
Distance to nearest shops	0.2691	0.3115
Distance to nearest restaurants	0.3488	0.3597
Distance to nearest public transport stop	0.2721	0.2035
Weather (extracted from the Royal Netherlands Meteorological Institute (KNMI))		
Temperature (in 0.1 °C)	236.64	45.830
Cloudiness (1–9)	4.35	3.402
Rain (Yes)	0.05	0.210

**Table 3 ijerph-16-03621-t003:** Goodness-of-fit of the model.

Degrees of Freedom	686
Full information Maximum-Likelihood Chi-square	553.58
Chi square/degrees of freedom	0.807
Root mean square error of approximation (RMSEA)	0.00
90% Confidence interval for RMSEA	0.00; 0.00
*P*-value for test of close fit (RMSEA < 0.05)	1.00

**Table 4 ijerph-16-03621-t004:** Results multilevel path model (unstandardized estimates).

From/to	Satisfaction Urban Public Space	Momentary SWB	Long-term SWB	
	Coefficients	Coefficients	Coefficients	
**Within level (momentary experience level)**				
Satisfaction urban public space		0.02 **		
Location – Leisure/ café/ bar/restaurant		0.30 **		
Location – Shop/mall		−0.20 *		
Location – Relocating/ on the move		−0.60 **		
Distance – shops (in km)	2.22 **	0.51 **		
Distance - public transport stop (in km)	3.20 **			
Time/day – weekend	1.14 **	0.29 **		
Error variance	44.31	0.65	3.91	
R^2^	0.032	0.20	0.17	
R^2^ reduced form	0.032	0.17	0.17	
**Between level (participant level)**				
Long-term SWB		0.03 **		
Extroversion			0.70 **	
Neuroticism			−1.65 **	
Homeownership			2.62 **	
Work situation – Retired			2.71 **	
Health – Reasonable/ Bad			−2.75 **	
Life changing event – moving			1.86 **	
Error variance	44.31	0.68	3.91	
R^2^	0.00	0.049	0.78	
R^2^ reduced form	0.00	0.016	0.78	

Note: * *p* < 0.05, ** *p* < 0.01.

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
