# Peer review of "A Multi-Level Path Analysis of the Relationships between the Momentary Experience Characteristics, Satisfaction with Urban Public Spaces, and Momentary- and Long-Term Subjective Wellbeing"

_ijerph, 2019, doi:10.3390/ijerph16193621_

Round 1
Reviewer 1 Report
Very well written.
i am a little concerned by the number of hypotheses:17 hypotheses
BELOW ARE JUST A FEW MINOR POINTS:
Line 121. CHANGE “Also, some studies…” to “However, some studies…”
Line 126 MISTAKE “Saw, Lim & [39] showed” YOU ARE MISSING Carrasco
Line 150 TYPO “Hypothesis 2i (H2i): Personality influence momentary and long-term SWB” SHOULD BE INFLUENCES
Line 181 “more active travel activities “ MAYBE CHANGE activities TO modes (JUST A SUGGESTION)
Line 380 TYPO “This relation was also found by Saw, Lim & Carrasco [38].” SHOULD BE [39]
Reviewer 2 Report
This study examined associations between momentary satisfaction with urban built environments and subjective wellbeing while controlling for individual and contextual characteristics. The manuscript possesses the potential to inform urban design standards that could theoretically advance population health and wellbeing, and the authors are commended for taking a novel approach to examining the complex relationship between built environments, momentary satisfaction, and subjective wellbeing. However, there appear to several limitations to the manuscript.
The introduction would benefit from clarification of key arguments. For example, on lines 61-61, the authors indicate that “many people being around (i.e., company) are more likely to have experiences.” While the authors intention seems apparent, the phrasing could be misleading. Do the authors mean that many people in a place promotes positive experiences? The differentiation between short- and long-term SWB is confusing as described in the introduction. This is clarified in the ensuing literature review. However, it would be helpful to make this clear upfront. The literature review tends to be overly schematic with insufficient evidence for key arguments. The second paragraph (lines 111 – 137) makes a case for controlling for individual-level characteristics, which is a valid argument. However, the authors introduce numerous variables—e.g., education, gender, age, marital status, etc. It would benefit the review to include additional references regarding these important variables, with additional analysis of their importance in SWB studies. Furthermore, a one paragraph review is not sufficient. Again, regarding the review pertaining to hypothesis 3, several variables are explained in single sentences, thus lacking sufficient explanation of their association with SWB. The literature review for the fourth hypothesis provides a much better developed explanation of how physical activity and travel behavior associate with SWB. This likely is a result of focusing on fewer variables. The review does not make a case for the importance of urban environments. That is, while the authors identify several urban environment features that associate with momentary satisfaction and SWB, many of which are found in suburban environments (e.g., public parks, cafes, etc.), it is unclear what is being measured that is uniquely urban. This should be clarified, as it is a central focus of the manuscript. Finally, SWB needs to be clarified and used uniformly throughout the review. There are too many instances of SWB without indication of momentary and long-term, thus making it difficult to identify how the authors interpreted the literature. The methods section would benefit from additional clarification. On line 224, the authors indicated that they administered an online questionnaire with open- and multiple-choice questions. However, the authors did not explain how open-choice questions were coded and analyzed. On lines 235-237, the authors contend “the chance that people will have a people will have a lot of negative of positive experience, with regard to the urban space, is limited…” is very confusing. Are they indicating that chances are higher in suburban space, or perhaps they mean to say that this is not possible in a finite time range? The authors also state that questionnaire respondents were asked to report positive and negative experiences in “urban public locations/places” (lines 239-240). However, it is unclear if respondents were provided a list of examples of these “urban” locations or if they simply reported experiences of any place. This could be methodologically flawed, particularly given the small sample size. The explanations of momentary and long-term SWB are much clearer in the methods section compared to the literature review. The authors report experience characteristics starting on line 295. However, up to this point in the manuscript, it is unclear how the investigators operationalized “momentary experience.” This should be clarified upfront. Please specify with version of LISREL was used in the main text. Given the stated sample size, which is quite small considering the number of variables, the authors should provide a detailed explanation of why path analysis is appropriate for this study. Finally, the manuscript would benefit from a detailed explanation of why the authors tested hypotheses based on what appears to be posteriori path analysis given the explanation of model development on lines 332-335. The results are relatively clear, but would benefit from inclusion of key explanatory factors. For example, the authors do not investigate collinearity, thus making it difficult to verify bias in the model. Furthermore, it’s challenging to assess the results absent adequate explanation of the methodological approach. Overall, the results seem to confirm SWB associations indicated in the literature review. Implications discussed throughout should be examined in greater depth. For example, on lines 377-378, the authors indicate that results could be used by policy makers, but they did not indicate how this data could be used. Furthermore, these implications should be in a discussion section. The conclusions would benefit from a grounding of what makes the “experiences” examined in this manuscript uniquely urban. In addition, the conclusions should examine study findings for public health. The only implication discussed concerns urban greening on lines 451-453. Thus, the paper would benefit from an examination of how study results may inform urban design for public health more specifically. Again, the uniqueness of what the urban context needs to be examined. It’s still not clear what uniquely urban places were included in the study. Finally, the authors indicate that this study provides new insights regarding experience of urban space. These new insights should be spelled out, with implications of each examined.
Reviewer 3 Report
This paper presents the results of an event sampling study regarding the relationship of elements of the urban environment and short-term subjective well-being. Strengths of this study include an innovative methodological approach and use of sophisticated analytical techniques. However, there are some significant limitations that I think need to be addressed.
The details of the ESM design are not fully explained, particularly with regards to the prompts that participants were given and to the specificity of the reporting method. It is stated that participants were given reminder prompts twice daily, but it is important also to give more specific details about the contents of the prompts (i.e., how was the type of event meant by “negative or positive experiences with regard to urban public locations” defined for participants) and how they were delivered (e.g., email, text?). It is also important to know more details about the average latency between the events and the reports, since the reporting of both event details and SWB outcomes are likely to become less precise as the time between the event and the report increase. The low rate of event-level responding is a serious concern may call the validity of the conclusions into question. Participants were prompted twice daily to enter experiences, yet the average number of events recorded was less than 0.5 per day – and it appears from the figure that a substantial number of participants reported only a single event over the entire 2 week study period. It seems unlikely that this is reflective of the actual rate with which participants experienced urban public spaces, so the apparently high rate of non-response at the event level needs to be addressed directly. Can you offer any insight into this phenomenon, or preferably conduct some checks regarding the robustness of the data to help support its validity. E.g., are there differences in the characteristics and/or SWB associated with events reported by regular responders (e.g., n >= 14), and very low responders (e.g., n = 1). It is important to consider whether there are differences in response quality between participants who were engaged with the task and those who may have done the minimum required to receive the study incentive. Even among those who did engage with the task, more consideration is needed of the issue of event-level response bias. People are more likely to notice and to take the time to report experiences that were more extreme (in terms of positivity/negativity) in comparison to relatively neutral ones. More importantly, there may be individual differences in terms of bias towards reporting positive or negative experiences (e.g., people with higher long-term SWB may be primed to respond to positive experiences, and those with low SWB may be primed to respond more to negative experiences). This can be partly assessed analytically, but should also be part of the limitations section. The model conceptualization is somewhat unclear in terms of why some of the event-level characteristics have paths directly to SWB and others indirectly via urban space satisfaction. Were these paths determined purely empirically through backward deletion, or are there theoretical reasons for structuring it this way? The use of backward stepwise deletion seems to raise the risk of leaving out important covariates, as well as increasing the risk of over-fitting the model. I think it would be advisable to at least offer a comparison with a “full” model, rather than just the version that has been pared down to significant paths. Most of the significant results remaining after stepwise deletion seem to be only nominally related to the stated aims of assessing effects of elements of urban space. It would seem that people would naturally tend to have higher SWB in leisure-related locations, and less when using transportation or in shopping locations, since leisure activities are inherently intended to improve SWB, and the others are not. Similarly, people would naturally tend to have higher short-term SWB on the weekend, because they are less likely to be working. More work needs to be done in the discussion to link these variables with the urban environment rather than with incidentally related aspects of the event.Author Response
Please see the attachment.

Reviewer 4 Report
The research focuses on a theme highly relevant for policy makers, planners, landscape architects, urban designers and architects: how do objective and subjective aspects of built environment influence SWB?
The current manuscript lacks a comprehensive coverage of the past literature or in-depth discussion of the intellectual debates and theoretical progress in the literature on the momentary satisfaction with urban spaces, momentary- and long-term SWB, as well as affective wellbeing and cognitive wellbeing. Theoretical thinking needs to be enhanced. The existing introduction and discussions on the current literature are too much descriptive. Sharp arguments are very important. What are the new theoretical arguments you proposed?
Rather than just simply listing if the hypotheses were confirmed (or not), more "your own" ideas or criticism are needed. It is important to delve more into the theoretical discussions underlying the empirical findings. The author needs to make it clearer how each concept / idea /result and theory are inter-related.
Please, present “your results” clearly. There is little discussion of the theoretical debates underlying the empirical findings. Please, make clear discussions and suggestions for future research agenda.
The author needs to make the contribution and significance of this paper more explicit.
What future research agenda do you suggest?
The author needs to make clearer how the results will guide the development of policy and design of “healthy, attractive, liveable and safe living environment for citizens”.
Round 2
Reviewer 2 Report
Thank you for addressing my comments.
Reviewer 3 Report
All of my comments regarding the original draft of this manuscript have been addressed.
Reviewer 4 Report
The discussion section and conclusion improved considerably. The methodology and some of the ideas (concepts and terms) expressed in the manuscript are clearer.